# Factors Associated with Underweight, Overweight, and Eating Disorders in Young Korean Women: A Population-Based Study

**DOI:** 10.3390/nu14061315

**Published:** 2022-03-21

**Authors:** Youl-Ri Kim, Zhen An, Kyung-Hee Kim, Da-Mee Kim, Bo-In Hwang, Mirihae Kim

**Affiliations:** 1Department of Neuropsychiatry, Seoul Paik Hospital, Inje University, Mareunnaero 9, Jung-gu, Seoul 04551, Korea; 2Institute of Eating Disorders and Mental Health, Inje University, Mareunnaero 9, Jung-gu, Seoul 04551, Korea; qtnvnxx@ucl.ac.uk (Z.A.); boinhh@gmail.com (B.-I.H.); 3Department of Food and Nutrition, Duksung Women’s University, 31 Samyang-ro, Dobong-gu, Seoul 01369, Korea; khkim@duksung.ac.kr (K.-H.K.); dameedamee@naver.com (D.-M.K.); 4Department of Psychology, Duksung Women’s University, 31 Samyang-ro, Dobong-gu, Seoul 01369, Korea; medehae@duksung.ac.kr

**Keywords:** eating behavior, overweight, obesity, underweight, eating disorders, anorexia nervosa, bulimia nervosa, nutrients

## Abstract

Both underweight (UW) and overweight (OW) conditions are problematic in young women. The aim of this study was to examine the factors associated with extreme weight status and eating disorders (EDs) in young Korean women. A total of 808 women (mean age 22.3 ± 3.4 years) participated, including 144 with UW [Body Mass Index (BMI) < 18.5kg/m^2^], 364 with NW, and 137 with OW or obesity (BMI ≥ 25kg/m^2^), and 63 patients with anorexia nervosa (AN) and 100 with bulimia nervosa (BN). Participants completed questionnaires regarding nutrients consumed, eating behaviors, health behaviors, body image, and obsessive-compulsive symptoms with face to face interviews. The associations between the status of participants and the data were analyzed with NW group as a reference. OW status was associated with overeating and with frequent eating. UW status was associated with less frequent overeating and with longer sleep duration. AN status was associated with less frequent consumption of alcohol. BN status was associated with a larger discrepancy between the ideal and current body shape. Both OW status and BN were associated with more obsessive-compulsive symptoms. The results suggested that certain dietary, health, and psychological factors are associated with extreme weight conditions and EDs.

## 1. Introduction

Both being underweight and being overweight are linked to a higher risk of negative health effects [1,2,3,4]. Together with the global increase in the rate of obesity, the high prevalence of young women who are underweight is problematic in low and middle-income countries as well as high-income countries [5,6,7]. In Korea, in the year 2009, the rates of underweight and overweight conditions in young women aged 19–29 were 16.8% and 14.3%, respectively, and the percentages were 15.2% and 16.5%, respectively, in the year 2019 [8]. The body mass index (BMI) of Korean female college students was the lowest at 19.3 kg/m^2^ according to the International Health Behavior Survey in 22 countries [9]. Female college students were reported to have chosen to lower their food consumption rather than increase their exercise due to a lack of nutritional information, resulting in undesirable weight reduction or eating disorders (EDs) [10]. Though the fundamental cause of weight gain is a long-term energy balance between calories consumed, the understanding of the etiological factors and the interaction between these factors is incomplete [11].

There is considerable evidence showing unhealthy eating patterns and body dissatisfaction to be risk factors for both EDs and obesity [12,13,14], but it has been disputed as to whether these factors are related to BMI [15,16]. The identified core features related to abnormal weight and EDs have been insufficient to characterize each group due to limited variables employed in the previous studies.

College years are associated with a significant reduction in students’ healthy lifestyle behaviors [17] and females, in particular, are at a higher risk of developing disordered eating behaviors [14]. College campuses are faced with an elevated prevalence of EDs, with 13.5% of US college women being screening positive [18]. In addition, the early years of college are a critical developmental window for weight gain [19]. Identifying the determinants of excessive weight and EDs in university students may thus help them to adopt healthy behaviors to manage risk factors that affect their later years.

The aims of this study were to cross-sectionally investigate the factors that bring about extreme weight status and EDs in young women. We used the BMI criteria for inclusion in underweight (UW), normal-weight (NW) and overweight or obesity (OW) groups to reflect the characteristics of college students with extreme weight conditions. Our first hypothesis was that extreme weights are not a quantitatively linear condition in regard to etiological factors. The second hypothesis was that anorexia nervosa (AN) and bulimia nervosa (BN) are not extreme conditions, but complicated conditions compared to UW and OW conditions, respectively. The etiological factors used in this study were health behaviors, nutrition, obsessive-compulsive symptoms and body image discrepancy. This is expected to facilitate the development of prevention and intervention plans for obesity and EDs in young women.

## 2. Materials and Methods

### 2.1. Participants

Female university students were recruited from August to December of 2016 via research advertisements on the social networking websites of 14 universities in Seoul metropolitan areas. Patients were recruited in the ED outpatient clinic of Seoul Paik Hospital, and included women with AN or BN of16 years of age or older. Parts of the study were published previously [20]. For the universal application of the results of the study, we used the strict international criteria of OW (BMI ≥ 25 kg/m^2^) [21] in classification of BMI instead of the Asia-Pacific regional guidelines (BMI ≥ 23.5 kg/m^2^) [22]. The BMI was calculated using weight and height measurements at recruitment, and university participants were classified into three groups: UW (BMI < 18.5kg/m^2^, *N* = 144), NW (18.5–24.9 kg/m^2^, *N* = 364), and OW (25–29.9 kg/m^2^, *N* = 110, and ≥30 kg/m^2^, *N* = 27), respectively. University students were included in the study only if they did not self-report ED diagnosis. Exclusion criteria were assessed via self-reports on: (1) severe physical illnesses related to abnormal weights requiring treatment (e.g., diabetes, thyroid problems, and cystic fibrosis, etc.); (2) substance use disorder or alcohol dependence; (3) learning difficulties; and (4) pregnancy. All participants were subjected to an interview and questionnaires.

As shown in Figure 1, a total of 808 young women participated in the study, which consisted of university students (364 participants with NW, 144 with UW, and 137 with OW) and 163 patients with ED (63 patients with AN and 100 patients with BN).

The mean age of the students was 21.96 (SD = 2.58) years in the NW group, 22.41 (SD = 2.72) years in the UW group, and 22.88 (SD = 3.27) years in the OW group. The mean age of the patients was 22.57 (SD = 5.47) years in the AN and 22.37 (SD = 4.78) years in the BN groups. Of the students, 614 (95.2%) were undergraduate students or had undergraduate degrees, 16 (2.5%) were graduate students or had graduate degrees, and 10 participants (1.5%) had high school diplomas or less. Of all the undergraduates, 105 (16.3%) were in their first year, 116 (18%) were in the second year, 167 (25.9%) were in the third year, 152 (23.6%) were in the fourth year, and five (0.8%) were in other years.

For the clinical participants, 18 participants (11.1%) had completed high school or less, 64 participants (39.2%) were current undergraduates or had undergraduate degrees, and 12 participants (7.4%) were graduate students or had graduate degrees.

The participants provided informed consent before participating. Parental consent was obtained for those who aged under 18 (*N* = 28), all of whom were patients with EDs. This study was approved by the Institutional Review Board (IRB) of Inje University (IRB No. INJE 2016-01-003-002).

### 2.2. Measures

Health-related behaviors. Health-related behaviors were assessed by interview questionnaires. The individual component interview questionnaire included information on the duration of sleep, frequency of drinking alcohol (frequency per month), smoking status (i.e., current smoker, ex-smoker, and non-smoker), and the frequency of exercise (days per intensity). Health interviews were conducted by trained staff members including physicians, psychologists, and health interviewers using face-to-face interviews and a self-administered questionnaire.

Nutrition survey. The nutrition survey addressed dietary behaviors and food intake for 24 h. The dietary behavior questionnaire included the frequency of overeating (frequency per week), regularity of mealtime (regular meal day per week), estimated time for a meal, and the amount of rice consumed per meal. The food intake questionnaire was an open-ended survey for reporting various dishes and foods using the 24-h recall method. Information was collected between Tuesday and Friday, and we checked the dietary intake for weekdays. If the day when the examination took place was special (e.g., birthday dinner or fasting for a medical check-up), we collected dietary data from the day before the planned day. A trained researcher conducted a face-to-face interview with the participant and used food models and the actual photograph of one portion of food to assess the exact amount of food consumed. The data from the recall method were analyzed using the Computer-aided Nutrition Analysis Program 5.0, Korean Nutrition Society (CAN PRO 5.0). The total amount of food, energy (kcal), carbohydrates, lipids, protein, and other nutrients consumed in a day was calculated. We calculated the amount of each nutrient consumed per 1000 kcal to measure the nutritional density of each nutrient.

Figure Rating Scale (FRS) [23]. Each participant was presented with a series of nine female silhouettes of increasing body sizes and asked to choose the ones that best represented their current body shape and their ideal body shape. Body image disturbance was calculated by subtracting the ideal body size from the current body size.

Obsessive-Compulsive Inventory-Revised (OCI-R) [24]. The OCI-R is an 18-item self-reported scale consisting of six subscales on symptoms of obsessive-compulsive disorder: washing, obsessing, mental neutralizing, ordering, hoarding, and doubting. The reliability of the Korean version of the OCI-R was high (Cronbach’s α = 0.90) [25].

### 2.3. Statistical Analysis

To examine the association between dietary behaviors, food intake, health behaviors, and psychological features, and the group (UW, OW, AN, and BN groups), we used the logistic regression model and the group as a response variable with the NW group as the reference. If an independent variable was categorical, the last category of the variable was used as a reference category in logistic regression analysis. Statistical significance was set at the 5% level (*p* < 0.05). We used two-sided *p*-values and a 95% confidence interval (CI). All data were analyzed by SPSS 23.0 statistics software (SPSS Inc., New York, NY, USA).

## 3. Results 

### 3.1. Age and BMI of the Participants

Table 1 shows the descriptive statistics of age and BMI of the NW, UW, OW, AN, and BN groups. The mean age of the participants was 22.30 years (SD = 3.36), with no significant difference between the groups (*p* = 0.076). BMI differed significantly between the groups (*p* < 0.001).

### 3.2. Dietary Behaviors and Food Intake

Table 2 shows the comparison of dietary behaviors and food intake in young women with UW or OW and patients with AN or BN compared to the young women with NW.

#### 3.2.1. Nutrient Intake

The UW group consumed larger proportions of carbohydrates [OR (95% CI) = 1.01 (1–1.02), *p* = 0.004] and fewer lipids per 1000 kcal compared to the NW group [OR (95% CI) = 0.98 (0.96–1.00), *p* = 0.044]. They consumed fewer kcal than the NW group [OR (95% CI) = 0.72 (0.53–0.99), *p* = 0.043]. In the OW group, there were no differences in kcal and the proportion of nutrients consumed compared to the NW group. The BN group consumed more kcal than the NW group [OR (95% CI) = 1.97 (1.39–2.79), *p* < 0.001]. Both AN and BN groups had higher cholesterol intakes compared to the NW group [OR (95% CI) = 1.01 (1.01–1.02), *p* < 0.001 for both].

#### 3.2.2. Dietary Behaviors including Overeating

The UW group was associated with less frequent overeating [OR (95% CI) = 0.30 (0.13–0.65), *p* = 0.003 for daily and 0.36 (0.2–0.6), *p* < 0.001 for weekly, respectively] compared to the NW group. Assuming that 10–30 min of mealtime per meal is appropriate, the UW group was less likely to spend <10 min for a meal compared to the NW group [OR (95% CI) = 0.09 (0.01–0.7), *p* = 0.021].

When one bowl of rice is assumed to be an appropriate portion per meal, the OW group was associated with consumption of more rice [OR (95% CI) = 3.21 (1.65–6.24), *p* = 0.001] and more frequent overeating than the NW group [OR (95% CI) = 4.58 (1.45–14.44), *p* = 0.009].

Having AN was associated with consumption of less rice at each meal [OR (95% CI) = 0.33 (0.16–0.74), *p* = 0.004] and the patients were more likely not to allow themselves to overeat even once a week [OR (95% CI) = 0.29 (0.13–0.61), *p* = 0.003] compared to the NW group. The AN group was more likely to spend over 30 min for a meal compared to the NW group [OR (95% CI) = 0.2 (0.09–0.43), *p* < 0.001]. Assuming that two to five days of regular mealtime per week is usual, the AN group was associated with more regular mealtime compared to the NW group [OR (95% CI) = 0.24 (0.1–0.58), *p* = 0.002].

### 3.3. Health Behaviors

The health behaviors included sleep, alcohol consumption, smoking, and exercise. Table 3 shows the health behaviors of young women with NW, UW or OW, and patients with AN and BN, along with the association between the behaviors and groups. For sleep duration, the UW group was associated with longer sleep compared to the NW group [OR (95% CI) = 1.24 (1.06–1.46), *p* = 0.006]. The frequency of drinking alcohol in the UW, OW, and BN groups was not significantly different from that in the NW group, but the AN group drank less frequently [OR (95% CI) = 0.2 (0.09–0.44), *p* < 0.001]. For smoking, both ED groups were more likely to be current smokers compared to the NW group [AN: OR (95% CI) = 4.33 (1.42–13.16), *p* = 0.01; BN: OR (95% CI) = 3.78 (1.25–11.45), *p* = 0.018], whereas women in the OW group were more likely to be former smokers [OR (95% CI) = 3.321 (1.49–7.39), *p* = 0.003]. The BN group performed more intense physical activities [OR (95% CI) = 1.36 (1.14–1.61), *p* = 0.001] and less mild physical activities (i.e., walking) compared to the NW group [OR (95% CI) = 0.8 (0.69–0.91), *p* = 0.001].

### 3.4. Psychological Factors

The psychological factors included obsessive-compulsive symptoms and body image discrepancy. Table 4 shows the psychological factors of young women with NW, UW or OW and patients with AN or BN, along with the association between the factors and the groups. Both BN and OW groups had more obsessive-compulsive symptoms compared to the NW group [all *p*-values < 0.001]. The AN group had higher scores on the doubting, obsessing, ordering, and neutralizing subscales than the NW group [OR (95% CI) = 1.13 (1.01–1.26), *p* = 0.026; 1.19 (1.06–1.33), *p* = 0.003; 1.11 (1.01–1.22), *p* = 0.036; and 1.23 (1.1–1.37), *p* < 0.0001, respectively]. Patients in the BN group perceived their current body shape to be bigger than the NW group did [OR (95% CI) = 2.09 (1.51–2.89), *p* < 0.001]. The ideal body shape in the BN group was smaller than that in the NW group [OR (95% CI) = 0.55 (0.35–0.85), *p* = 0.007]. The discrepancies between the current and ideal shapes were smaller in the UW and AN groups (both *p*-values < 0.001) and larger in the OW and BN groups (both *p*-values < 0.001) compared to the NW group.

## 4. Discussion

In this study, we examined the associations between health behaviors, nutrition, psychological factors, and extreme weight conditions and EDs in young women. Women with UW consumed more carbohydrates with lower lipid intake per 1000 kcal, overate less frequently, ate meals less quickly, and had longer sleep durations. Being OW was associated with overeating more frequently and being a former smoker. The women in both ED groups consumed higher proportions of cholesterol per 1000 kcal, and women with BN had the highest total caloric intake. Women with AN ate for longer periods of time, had stricter mealtimes, and consumed less alcohol. The OW status and BN were associated with higher obsessive-compulsivity scores. Women with BN exhibited the greatest gap between their ideal and perceptual current shape. EDs are complicated conditions rather than the extremes of a normal condition.

The results support that energy homeostasis is a core factor that maintains extreme weights status. Women with OW more frequently overate rather than consuming more total kcal, as previous research has revealed the coexistence of binge eating behaviors in young women with OW [26]. It was also noted that women with UW were less likely to overeat or eat quickly, reflecting a stable and moderate diet style. A previous Polish study that reported similar results found that female students with UW skipped meals more rarely than the OW students [27]. These findings show that the maintenance of low weight was associated with balanced and stable diet behaviors. Meanwhile, for the energy intake, the Korean women with NW had an intake of 1590 kcal per day, which was lower than the estimated energy requirement for women aged 19–29 years [28].

Patients with AN had more regimented mealtimes. Patients with BN exhibited extreme eating behaviors in which they were more likely to consume too much or too little, rather than an adequate amount. Women with UW slept for longer periods of time. This might be explained by the decreased ghrelin and increased leptin levels accompanied by low BMI in contrast to the results reported in obesity [29]. Lower alcohol consumption in women with AN could be explained by their fear of gaining weight due to the high calories in alcoholic drinks or by their highly self-regulative features. Women with BN tended to have extreme exercise patterns, as evidenced by their excessive exercise and low frequency of light exercise. The levels of obsessive-compulsive symptoms were higher in women with OW as well as in the ED groups. These findings reflected their general obsessive-compulsive traits in addition to their high food-specific obsessions and compulsivity related to food [30].

Discrepancies between the ideal and current body shape were prevalent in women with OW and BN, but not in those with UW and AN, when it came to body image. Our results are consistent with previous studies that found greater body dissatisfaction in patients with BN than patients with AN [31], and the direct association of BMI with body dissatisfaction in female university students [32]. In our study, patients with BN sought the thinnest body shape among the groups but perceived their body to be bigger than its actual size. The visual and perceptual disturbances in patients with BN might be the driving force of their vicious dieting cycles.

The strength of this study was the inclusion of adequate samples of young women with extreme weights on the basis of the WHO global criteria [21]. There are a few limitations that need to be considered in the interpretation of the study. The first is that we employed a brief self-report instead of a screening tool to exclude diagnoses of EDs from the university students. The second is that most of the patients with EDs were undergoing nutritional counseling, and thus their nutrient intake and eating behaviors might be somewhat corrected. The third is that the university students and the patients were recruited from different populations. However, we do not believe it caused biased results, as the groups of individuals represented their general population.

## 5. Conclusions

In conclusion, our results demonstrated differences in nutrients, dietary and health behaviors, and psychological factors among young women with UW, OW, AN or BN compared to women with NW. There were differences between the young women with OW and young women with UW in the balance in their dietary styles. In contrast, women with UW reported more homeostatic eating behaviors, and women with OW and patients with EDs were on the extremes of eating behaviors. The patients with BN had discrepancies between their current and ideal body shapes.

It is expected that the findings of this study will contribute to improving prevention and interventions for young women with obesity and EDs.

## Figures and Tables

**Figure 1 nutrients-14-01315-f001:**
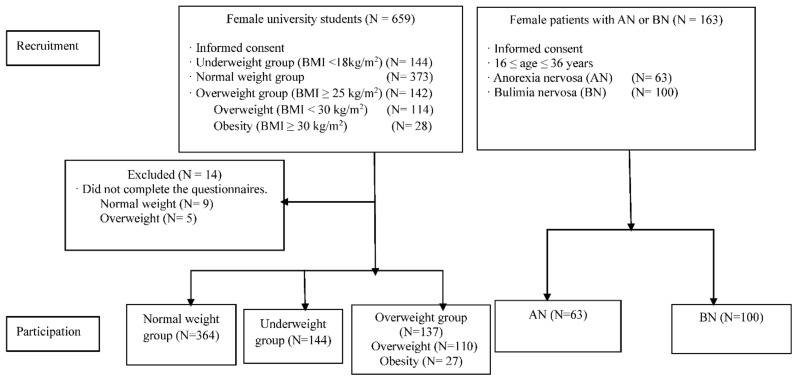
Flow chart of study participants. AN: anorexia nervosa; BN: bulimia nervosa.

**Table 1 nutrients-14-01315-t001:** Age and BMI of young women with NW, UW or OW, and patients with AN and BN.

	NW (*n* = 364)	UW (*n* = 144)	OW (*n* = 137)	AN (*n* = 63)	BN (*n* = 100)	ANOVA		
						*F*	*df*	*p*
Age, year	21.96 (2.58)	22.41 (2.72)	22.88 (3.27)	22.57 (5.47)	22.43 (4.74)	2.12	4803	0.076
BMI, kg/m^2^	21 (1.59)	17.3 (0.78)	28.1 (3.37)	15.57 (2.37)	21.21 (2.93)	573.17	4798	<0.001

NW, Normal-Weight (18.5 ≤ BMI < 25 kg/m^2^); UW, Underweight (BMI < 18.5 kg/m^2^); OW, Overweight (BMI ≥ 25 kg/m^2^); AN, Anorexia Nervosa; BN, Bulimia Nervosa. Data are shown as mean (SD).

**Table 2 nutrients-14-01315-t002:** Nutrient intake and dietary habits of young women with UW and OW, and patients with AN and BN compared to women with NW.

	NW (*n* = 364)	UW (*n* = 144)	OW (*n* = 137)	AN (*n* = 63)	BN (*n* = 100)	NW vs. UW		NW vs. OW		NW vs. AN		NW vs. BN	
						Exp(B) (95% CI)	*p*	Exp(B) (95% CI)	*p*	Exp(B) (95% CI)	*p*	Exp(B) (95% CI)	*p*
Nutrient intake													
Intake energy (1000kcal/day)	1.59 (0.61)	1.47 (0.53)	1.72 (0.65)	1.49 (1.16)	2.07 (1.28)	0.72 (0.53–0.99)	0.043 *	1.28 (0.97–1.69)	0.081	0.78 (0.47–1.29)	0.33	1.97 (1.39–2.79)	<0.001 ***
Carbohydrate (g/1000kcal)	132.68 (28.84)	141.1 (27.36)	132.53 (32.08)	139.09 (34.34)	130.87 (31.45)	1.01 (1.00–1.02)	0.004 **	1.00 (0.99–1.01)	0.96	1.01 (1.00–1.02)	0.18	1.00 (0.99–1.01)	0.70
Lipid (g/1000kcal)	33.26 (10.21)	31.2 (9.74)	33.75 (10.92)	31.85 (11.72)	35.66 (12.4)	0.98 (0.96–1.00)	0.044 *	1.00 (0.99–1.02)	0.65	0.99 (0.96–1.02)	0.40	1.02 (0.99–1.05)	0.15
Protein (g/1000kcal)	37.34 (10.4)	35.47 (9.23)	38.25 (10.53)	41.47 (17.11)	36.41 (13.32)	0.98 (0.96–1.00)	0.073	1.01 (0.99–1.03)	0.41	1.03 (1–1.05)	0.024 *	0.99 (0.96–1.02)	0.58
Cholesterol (mg/1000ckal)	42.51 (53.8)	38.3 (39.47)	46.03 (68.58)	208.16 (244.03)	223.19 (254.67)	1.00 (0.99–1.00)	0.34	1.0 0 (1.00–1.01)	0.46	1.01 (1.01–1.02)	<0.001 ***	1.01 (1.01–1.02)	<0.001 ***
Amount of rice/meal (Bowl(s) of rice)													
≥1	156 (43.21%)	70 (48.95%)	79 (58.09%)	13 (29.55%)	23 (46.94%)	1.07 (0.65–1.76)	0.80	3.21 (1.65–6.24)	0.001 **	0.33 (0.16–0.74)	0.004 **	0.62 (0.32–1.22)	0.17
2/3	129 (35.73%)	41 (28.67%)	45 (33.09%)	12 (27.27%)	8 (16.33%)	0.76 (0.44–1.30)	0.31	2.21 (1.1–4.44)	0.026 *	0.37 (0.17–0.81)	0.013 *	0.26 (0.11–0.63)	0.003 **
≤1/2	76 (21.05%)	32 (22.37%)	12 (8.82%)	19 (43.18%)	18 (36.73%)	1.00		1.00		1.00		1.00	
Time required for a meal													
<10 min	19 (5.26%)	1 (0.7%)	10 (7.35%)	6 (13.33%)	8 (16.33%)	0.09 (0.01–0.7)	0.021 *	2.04 (0.69–6.07)	0.20	0.75 (0.25–2.32)	0.62	2.61 (0.74–9.16)	0.13
10–30mins	311 (86.15%)	123 (86.01%)	118 (86.76%)	26 (57.78%)	36 (73.47%)	0.65 (0.35–1.19)	0.16	1.47 (0.66–3.29)	0.35	0.2 (0.09–0.43)	<0.001 ***	0.72 (0.26–1.96)	0.52
>30mins	31 (8.59%)	19 (13.29%)	8 (5.88%)	13 (28.89%)	5 (10.2%)	1.00		1.00		1.00		1.00	
Regularity of meal times													
≤1/week	8 (2.22%)	13 (9.09%)	9 (6.62%)	10 (22.73%)	5 (10.2%)	2.44 (0.82–7.21)	0.11	1.50 (0.48–4.65)	0.48	3.75 (1.1–12.79)	0.035 *	2.14 (0.53–8.68)	0.29
2–5/week	329 (91.14%)	114 (79.72%)	109 (80.15%)	26 (59.09%)	37 (75.51%)	0.52 (0.27–1.01)	0.055	0.44 (0.23–0.85)	0.014 *	0.24 (0.1–0.58)	0.002 **	0.39 (0.16–0.96)	0.04 *
≥6/week	24 (6.65%)	16 (11.19%)	18 (13.24%)	8 (18.18%)	7 (14.29%)	1.00		1.00		1.00		1.00	
Frequency of overeating													
≥1/day	44 (12.19%)	13 (9.09%)	26 (19.12%)	8 (17.78%)	18 (36.73%)	0.30 (0.13–0.65)	0.003 **	4.58 (1.45–14.44)	0.009 **	0.56 (0.2–1.59)	0.28	6.34 (1.31–29.33)	0.018 *
≥1/week	286 (79.22%)	99 (69.23%)	106 (77.94%)	27 (60%)	29 (59.18%)	0.36 (0.20–0.60)	<0.001 ***	2.87 (0.99–8.33)	0.052	0.29 (0.13–0.66)	0.003 **	1.57 (0.36–6.91)	0.55
None	31 (8.59%)	31 (21.68%)	4 (2.94%)	10 (22.22%)	2 (4.09%)	1.00		1.00		1.00		1.00	

NW, Normal-Weight (18.5 ≤ BMI < 25 kg/m^2^); UW, Underweight (BMI < 18.5 kg/m^2^); OW, Overweight (BMI ≥ 25 kg/m^2^); AN, Anorexia Nervosa; BN, Bulimia Nervosa Dates are shown as mean (SD), or frequency (%). Analysis by multinomial regression. The number of missing data was subtracted from the denominator when calculating the frequency (%). * *p* < 0.05, ** *p* < 0.01, *** *p* < 0.001.

**Table 3 nutrients-14-01315-t003:** Health behaviors in young women with UW and OW, and patients with AN and BN compared to women with NW.

	NW (*n* = 364)	UW (*n* = 144)	OW (*n* = 137)	AN (*n* = 63)	BN (*n* = 100)	NW vs. UW		NW vs. OW		NW vs. AN		NW vs. BN	
						Exp(B)	*p*	Exp(B)	*p*	Exp(B)	*p*	Exp(B)	*p*
Duration of sleep, h/day	6.49 (1.11)	6.83 (1.27)	6.66 (1.37)	6.49 (1.78)	6.79 (1.31)	1.24 (1.06–1.46)	0.006 **	1.11 (0.95–1.31)	0.19	1.00 (0.78–1.28)	0.98	1.21 (0.95–1.54)	0.12
Frequency of drinking													
≥2/week	45 (12.5%)	8 (5.56%)	10 (7.35%)	6 (13.95%)	10 (20.41%)	0.48 (0.21–1.09)	0.08	0.51 (0.24–1.08)	0.079	0.58 (0.22–1.48)	0.25	1.42 (0.61–3.27)	0.42
1–4/month	194 (53.89%)	91 (63.19%)	73 (53.68%)	9 (20.93%)	20 (40.81%)	1.27 (0.83–1.94)	0.28	0.86 (0.56–1.31)	0.48	0.20 (0.09–0.44)	<0.001 ***	0.66 (0.34–1.28)	0.22
<1/month	121 (33.61%)	45 (31.25%)	53 (38.97%)	28 (65.12%)	19 (38.78%)	1.00		1.00		1.00		1.00	
Frequency of smoking													
Smoker	11 (3.09%)	6 (4.17%)	5 (3.68%)	5 (11.63%)	5 (10.42%)	1.38 (0.50–3.80)	0.54	1.29 (0.44–3.80)	0.64	4.33 (1.42–13.16)	0.01 *	3.78 (1.25–11.45)	0.018 *
Ex-smoker	12 (3.37%)	6 (4.17%)	14 (10.29%)	3 (6.97%)	3 (6.25%)	1.26 (0.46–3.43)	0.65	3.32 (1.49–7.39)	0.003 **	2.38 (0.64–8.84)	0.20	2.08 (0.56–7.69)	0.27
Non-Smoker	333 (93.54%)	132 (91.66%)	117 (86.03%)	35 (81.4%)	40 (83.33%)	1.00		1.00		1.00		1.00	
Exercise, days													
Severe level	0.58 (1.2)	0.44 (1.05)	0.85 (1.45)	0.51 (1.37)	1.38 (2.37)	0.90 (0.75–1.09)	0.28	1.15 (1.00–1.33)	0.05	0.96 (0.71–1.29)	0.77	1.36 (1.14–1.61)	0.001 ***
Moderate level	2.01 (2.16)	2.16 (2.52)	2.12 (2.23)	1.44 (1.76)	1.67 (2.06)	1.03 (0.94–1.12)	0.52	1.02 (0.93–1.12)	0.66	0.87 (0.74–1.03)	0.12	0.93 (0.79–1.08)	0.34
Walking	5.82 (1.78)	5.71 (1.82)	5.74 (1.84)	5.4 (2.21)	4.85 (2.45)	0.97 (0.87–1.08)	0.54	0.98 (0.87–1.09)	0.65	0.89 (0.76–1.04)	0.15	0.8 (0.69–0.91)	0.001 ***

NW, Normal-Weight (18.5 ≤ BMI < 25 kg/m^2^); UW, Underweight (BMI < 18.5 kg/m^2^); OW, Overweight (BMI ≥ 25 kg/m^2^); AN, Anorexia Nervosa; BN, Bulimia Nervosa Data are shown as mean (SD), or frequency (%). Analysis by multinomial regression. The number of missing data was subtracted from the denominator when calculating the frequency (%). * *p* < 0.05, ** *p* < 0.01, *** *p* < 0.001.

**Table 4 nutrients-14-01315-t004:** Obsessive-compulsive symptoms and body image discrepancy in young women with UW and OW, and patients with AN and BN compared to women with NW.

	NW (*n* = 364)	UW (*n* = 144)	OW (*n* = 137)	AN (*n* = 63)	BN (*n* = 100)	NW vs. UW		NW vs. OW		NW vs. AN		NW vs. BN	
						Exp(B) (95% CI)	*p*	Exp(B) (95% CI)	*p*	Exp(B) (95% CI)	*p*	Exp(B) (95% CI)	*p*
OCI-R													
Washing	1.47 (1.99)	1.65 (2.25)	1.69 (2.24)	1.78 (2.54)	1.98 (2.65)	1.04 (0.95–1.14)	0.38	1.05 (0.96–1.15)	0.30	1.07 (0.93–1.22)	0.35	1.11 (0.98–1.25)	0.12
Doubting	2.47 (2.45)	2.74 (2.65)	3.49 (3.13)	3.38 (3.19)	3.81 (3.16)	1.04 (0.97–1.12)	0.28	1.14 (1.07–1.23)	<0.001 ***	1.13 (1.01–1.26)	0.026 *	1.18 (1.07–1.31)	0.001 **
Obsessing	1.7 (2.18)	1.85 (2.42)	2.38 (2.58)	2.82 (3.37)	3.18 (2.87)	1.03 (0.95–1.12)	0.49	1.12 (1.04–1.21)	0.004 **	1.19 (1.06–1.33)	0.003 **	1.24 (1.11–1.37)	0.001 **
Neutralizing	2.58 (2.55)	2.64 (2.44)	3.4 (2.84)	4.18 (3.02)	4.6 (3.11)	1.01 (0.94–1.09)	0.82	1.12 (1.04–1.21)	0.002 **	1.23 (1.1–1.37)	<0.0001 ***	1.31 (1.18–1.45)	<0.001 ***
Ordering	3.37 (3)	3.36 (2.98)	3.89 (2.96)	4.38 (3.9)	4.74 (3.35)	1.00 (0.94–1.07)	0.97	1.06 (0.99–1.13)	0.089	1.11 (1.01–1.22)	0.036	1.14 (1.04–1.25)	0.004 **
Hoarding	3.5 (2.71)	3.14 (2.76)	4.45 (3.14)	2.96 (3.04)	5.37 (0.61)	1.00 (0.89–1.02)	0.19	1.11 (1.04–1.19)	0.002 **	0.93 (0.83–1.05)	0.22	1.22 (1.11–1.34)	<0.001 ***
Total	15.09 (10.47)	15.37 (11.87)	19.29 (12.75)	19.49 (14.73)	23.84 (14.77)	1.00 (0.99–1.02)	0.79	1.03 (1.1–1.05)	<0.001 ***	1.03 (1.01–1.06)	0.014 *	1.06 (1.03–1.08)	<0.001 ***
FRS													
Current	4.18 (0.91)	2.78 (0.94)	5.97 (1.02)	2.78 (1.36)	4.88 (1.25)	0.24 (0.19–0.32)	<0.001 ***	6.28 (4.55–8.68)	<0.001 ***	0.24 (0.17–0.35)	<0.001 ***	2.09 (1.51–2.89)	<0.001 ***
Ideal	2.91 (0.66)	2.85 (0.72)	3.4 (0.71)	2.67 (0.74)	2.62 (0.73)	0.89 (0.67–1.17)	0.40	2.81 (2.06–3.81)	<0.001 ***	0.60 (0.38–0.94)	0.027 *	0.55 (0.35–0.85)	0.007 **
Body image Disturbance	1.27 (0.96)	−0.07 (1.16)	2.57 (1.09)	0.11 (1.47)	2.26 (1.25)	0.33 (0.27–0.41)	<0.001 ***	3.62 (2.78–4.1)	<0.001 ***	0.38 (0.28–0.51)	<0.001 ***	2.82 (2.04–3.90)	<0.001 ***

NW, Normal-Weight (18.5 ≤ BMI < 25 kg/m^2^); UW, Underweight (BMI < 18.5 kg/m^2^); OW, Overweight (BMI ≥ 25 kg/m^2^); AN, Anorexia Nervosa; BN, Bulimia Nervosa; OCI-R, Obsessive-Compulsive Inventory Revised; NEO-FFI, NEO Five Factor Inventory; FRS, Figure Rating Scales Data are shown as mean (SD); Analysis by multinomial regression. * *p* < 0.05, ** *p* < 0.01, *** *p* < 0.001.

## Data Availability

All data will be available upon requested.

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
