# Peer review of "Factors Associated with Underweight, Overweight, and Eating Disorders in Young Korean Women: A Population-Based Study"

_nutrients, 2022, doi:10.3390/nu14061315_

Round 1

Reviewer 1 Report

this is a nice and well described study on eating disorders in a population of Korean women. The topic is very interesting. The background section is lacking of important data on the psychiatric evaluation and follow-up. Also new references need to be included.

in particular: DOI: 10.1708/3104.30935; doi: 10.1007/s40519-020-01068-2

Reviewer 2 Report

The aim of the study was to examine the factors associated with extreme WS and ED in young Korean women. 808 women participated: 144 UW; 364 NW and 137 OW-OB. Sample also included 63 patients with AN and 100 with BN.

Participants completed questionnaires regarding nutrients consumed, eating behaviors, health behaviors, body image, and obsessive-compulsive symptoms with face-to-face interview.

Abstract OK

Introduction

Authors introduces COVID-19 lockdown as an important factor which had a significant impact on EB, increased weight gain and ED in some young adults. Furthermore, they stand the effect of lockdown increased weight concerns on extreme weight status. But they said no more about it and its relationship with the aim of the study. In the next sentence they described rates of UW and OW in 19-29 years old Korean young women. I detect lack of coherence.

Authors should better introduce the topic. Why the effect of lockdown matters if recruiting process was in 2016 (see Methods)? What happen with young women in Korea during this period? Why is relevant to the study? Perhaps they should describe better who are those “some young adults” (p1L38).

P2L50-53 “Additionally, most studies on abnormal weight employed the continuous variable of weight due to the low prevalence of extreme weights in young women, which could not reflect the characteristics of extreme weights”. I did not see any reference about it.

In general, authors should expand their introduction to cover relevant studies about the matter.

Materials and Methods.

I do not understand why the aim of the study is repeated starting this section. It is redundant.

Because many individuals with subclinical symptoms of ED are NW I wonder how authors controlled this issue. I did not see any exclusion criteria. [I already found a limitation about it into Discussion].

Results OK

Discussion needs to be improved.

Authors evaluate many aspects of EB, WS, EH and ED. They need to discuss more carefully every finding. They should use more relevant references to support (or not) their findings. Perhaps highlighting cultural similitudes and differences with other Asian and Western young women. It is just a suggestion, not mandatory.
